# Design of a Composite Based on Polyamide Fabric-Hydrogel-Zinc Oxide Particles to Act as Adsorbent and Photocatalyst

**DOI:** 10.3390/ma15196649

**Published:** 2022-09-25

**Authors:** Daniela Atanasova, Miglena Irikova, Desislava Staneva, Ivo Grabchev

**Affiliations:** 1Department of Textile, Leader and Fuels, University of Chemical Technology and Metallurgy, 1756 Sofia, Bulgaria; 2Faculty of Medicine, Sofia University St. “Kliment Ohridski”, 1407 Sofia, Bulgaria

**Keywords:** zinc oxide particles, textile composite, hydrogel, photocatalytic activity, adsorbent

## Abstract

Surface-initiated photopolymerization has been run to synthesize a hydrogel with ZnO particles distributed uniformly along its structure, which has been loaded onto a polyamide fabric. Three samples have been obtained at different concentrations of zinc nitrate (10% (sample PA10); 20% (sample PA20) and 30% (sample PA30) of the weight of the fabric, respectively)) and subjected to gravimetric analysis, scanning electron microscopy and transmission electron microscopy. The effect of the adsorption parameters of the composite material on the removal Drimaren Rot K-7B dye from water has been studied. The Freundlich isotherm describes this process better than the Langmuir isotherm. As the results of the adsorption kinetics show, the process fits well with a pseudo-second-order equation and depends both on the boundary layer and on the structure of the adsorbent itself. The thermodynamic parameters have demonstrated that the process is endothermic and physical. When exposed to ultraviolet light, the discoloration of the dye solution accelerates due to the photocatalytic properties of the composite materials. The addition of H_2_O_2_ also speeds up further the process, while the reuse of the materials slows it down, gradually changing the kinetic parameters. The reaction has been attributed to first-order kinetic model, when the active centers of the materials and the number of oxidative radicals formed are numerous and to the second-order kinetic model at a lower reaction activity. Moreover, 52% decolorization of the dye solution (50 mg L^−1^) in the dark was achieved from composite material PA 30 (13.3 g L^−1^) in 120 min and 89% under UV light irradiation. The H_2_O_2_ addition (0.14 mmol L^−1^) enhanced it up to 98%. In the second and third use of the photocatalyst, the dye removal decreased to 80% and 60%. Composite material PA30 exhibits antibacterial activity against Gram-negative bacteria *E. coli*, being most effective at eliminating Gram-positive bacteria *S. aureus.*

## 1. Introduction

Nowadays, wastewater purification is of crucial industrial and environmental significance worldwide. A lack of clean drinking water and contamination of water basins and groundwater with various chemicals (dyes, detergents, plant protection products, pharmaceuticals, plastics, synthetic products, etc.) represent the cause of the deterioration of ecosystems and human quality of life. One of the drawbacks of the widely applied technologies for dyeing textile materials is wastewater. Even when the amount of dye is small, the water is visibly colored, unlike other colorless pollutants. A fundamental requirement for textile dyes is to be stable under their application and use [1]. Drimarene K dyes are used widely in the textile industry. They are reactive dyes for dyeing fabrics, such as cotton, linen, and rayon. These modern reactive dyes are more stable in dry and liquid form than other reactive dyes.

Searching for effective methods for water purification, researchers have turned their attention to ZnO nanoparticles. Their conductive, piezoelectric, optical, and sensing properties open possibilities for vanguard industrial applications, including ultraviolet filters and photocatalysts [2,3,4,5]. In addition to the latter features, ZnO nanoparticles exhibit high antibacterial and antifungal properties [6,7]. Many studies report that reactive oxygen species (ROS) formation (including hydrogen peroxide H_2_O_2_, hydroxyl radicals OH^●^, and superoxide anion O_2_^●−^) is the major factor responsible for the antimicrobial effect of the ZnO nanoparticles and the destruction of chemical pollution (dyes, products of the pharmaceutical industry, etc.) [8,9,10]. These processes result in the mineralization of pollutants into inorganic ions and water. ZnO has been applied in powder form or as a thin film for wastewater treatment. In powder form, ZnO exhibits higher photo efficiency due to its larger active surface. Disadvantages of such an application are the possible aggregation of particles and the difficulty in their separating and regenerating. On the contrary, the thin film of ZnO can be separated easily, but its smaller active surface makes it less effective [11,12]. Therefore, the immobilization of ZnO nanostructures without agglomeration on a suitable substrate combines the advantages of a large surface area, easy regeneration, and reusability. This integration of adsorption with photocatalysis accelerates pollutant removal with mineralization. It also overcomes other shortcomings of individual materials, such as low adsorption and fast recombination of the photogenerated electrons and holes [13].

The design principles and modification methods are essential for effective heterogeneous photocatalytic materials development. Their effectiveness depends on factors, such as nanoparticle agglomeration and their interaction with the light, the available surface reaction sites, and the interfacial charge separation. The finding of optimal adsorption thermodynamics and kinetics is also valuable [14,15].

Textile materials have been investigated relatively little as integrated photocatalytic adsorbents, but they possess many advantages for these applications. The diversity in their composition and structural geometry defines their effectiveness and makes them a suitable matrix for obtaining composites with improved characteristics compared to other materials. The fibre materials are flexible, light, and durable, which determines their ease of use and maintenance. They have a large contact surface and are adaptable to uneven surfaces. The modification of textiles with hydrogel creates additional advantages. The hydrogel is a three-dimensional network and adsorbs contaminants into its highly porous structure. Their capability to swell is responsible for increased adsorption capacity compared with other adsorbents [16].

Recently, we have reported the synthesis of a hydrogel with iron oxide particles on a polyamide fabric by surface-initiated photopolymerization [17]. All composite materials have been investigated as catalysts of the heterogeneous Fenton process for treatment of real industrial wastewater after dyeing cotton fabric with the reactive dye Drimarene K-7B. A disadvantage of the Fenton process is the need for low pH values, H_2_O_2_ addition, and the possible iron leaching from the oxide surface. When zinc oxide is used, no additional treatment of the contaminated water is required, which accelerates the removal of water pollutants.

This study aims at producing a composite of polyamide fabric coated with a hydrogel containing ZnO nanoparticles so that the material can be applied for the adsorption and photodegradation of reactive dye Drimarene K-7B. The adsorption mechanism has been proven by determining the isotherm, kinetic models, and thermodynamic parameters. The factors, including the concentration of zinc ions as precursors of ZnO particles and the addition of hydrogen peroxide to the rate of dye photocatalytic degradation, have been determined. The kinetic study has demonstrated the photodegradation process and the material’s reusability. The antibacterial activity of the composite material has been tested against Gram-negative and Gram-positive bacteria.

## 2. Materials and Methods

### 2.1. Materials

A 100% polyamide fabric sample with a surface weight of 111 g m^−2^, satin-woven was used throughout the work. Acryl amide, (Fluka AG, Buchs, Switzerland); N,N′-methylenebisacrylamide (bis-AAm) (Sigma Aldrich, Darmstadt, Germany), Zn(NO_3_)_2_ × 6H_2_O (Sigma Aldrich) and N-methyldiethanolamine (MDEA) (Sigma Aldrich) were used without further purification as obtained. The reactive dye Drimaren Red K-7B (Clariant, Basel, Switzerland) was used for the preparation of the model dye solutions. Modified Eosin Y (MEY) with chloroacetyl chloride was synthesized according to a previously presented procedure [18]. All solutions were made with distilled water.

### 2.2. Preparation of Composite Materials ZnO Nanoparticles-Hydrogel-Polyamide Fabric

The modification procedure for polyamide fabric involved: dyeing with the modified Eosin Y; impregnation with an aqueous solution of Zn(NO_3_)_2_ at different concentrations; the synthesis of hydrogel on the fiber surface by surface-initiated photopolymerization with visible light and the formation of ZnO particles. The hydrogel and ZnO nanoparticle synthesis procedure was described in our previous study for cotton fabric modification [9]. It is summarized in Figure 1.

The dyed fabric with MEY was impregnated with an aqueous solution of zinc nitrate at different concentrations (10% (sample PA10); 20% (sample PA20) and 30% (sample PA30) of the weight of the fabric (owf), respectively)). Then the samples were dried in an oven at 80 °C for 10 min. Next, each sample was overlaid with an aqueous solution of acrylamide (25% owf), crosslinker N,N′-methylenebisacrylamide (bis-AAm) (5.0 wt % to acrylamide) and MDEA (50% to acrylamide, bis-AAm and Zn(NO_3_)_2_). The starting volume of solutions in each step was of liquor-to-goods ratio 1.7:1. The samples were illuminated with a lamp, emitting visible light for 5 h. Then, the samples were dried in an oven at 80 °C for 30 min and annealed at 140 °C for 3 min in air. The energy saving lamp that served as a photo-polymerization light source was HL 8325, 25 w, 1230 Lumen, 6400 K, Horoz Electric.

### 2.3. Characterization of the Prepared Composite Materials

#### 2.3.1. Gravimetric Analysis

The photopolymerization was evaluated by determination of the gel fraction. The samples of composite materials were measured in the dried state after photopolymerization and drying in the oven. They were soaked in distilled water for 18 h up to a constant weight and taken out from water in order to remove the soluble parts. The samples were dried again to constant weight and measured. The gel fraction percentage was calculated by the following Equation (1):(1)$$\mathrm{Gel}\;\mathrm{fraction}\;\left(\%\right)=\frac{W_{1}}{W_{2}}$$
where W_2_ and W_1_ are the weights of composite materials in the dry state before and after soaking in distilled water, respectively.

The weight gain percentage of the fabric after modification with hydrogel was calculated by Equation (2), where W_0_ and W are weights of fabric before and after treatment, respectively.
(2)$$\mathrm{Weight}\;\mathrm{gain}\;\left(\%\right)\;=\;\frac{\left(W-W_{0}\right)}{W_{0}}\times 100$$

#### 2.3.2. Structural Characterization

The modification of polyamide fabrics with hydrogel and ZnO nanoparticles was analyzed on a scanning electron microscope (SEM) JSM-5510 (JEOL, Tokyo, Japan), operated at 10 kV of acceleration voltage. The samples were coated with gold by JFC-1200 fine coater (JEOL, Tokyo, Japan) before imaging. To determine the morphology, size, and distribution of the zinc oxide particles in the hydrogel deposited onto the fibre surface a JEOL JEM 21,000, Tokyo, Japan transmission electron microscope (TEM) was used.

### 2.4. Study of the Obtained Material as an Adsorbent and Photocatalyst in the Decolorization of Solutions of Reactive Dye Drimaren Rot K-7B

The adsorption and photocatalytic effectiveness of the composite materials for the removal and decomposing of the model dye (reactive dye Drimaren Rot K-7B) were evaluated by spectrophotometric analysis. The absorption decrease of dye solution in the presence of composite materials was monitored by measuring samples at 550 nm with an ONDA UV-31 SCAN, spectrophotometer, 190 ÷ 1100 nm.

#### 2.4.1. Adsorption Experiment

The adsorption experiments were performed in the dark. The sample PA30 was immersed into a 20 mL solution of the dye in distilled water. Aliquots were taken at various time intervals and returned in the same vessel after measurement so that the liquid volume was kept constant. The adsorption capacity of sample was determined taking into account the initial and final solution absorption. The amount of adsorbed dye (mg g^−1^) was calculated by Equation (3):(3)$$q=\frac{\left(\mathrm{Co}-\mathrm{Ceq}\right)\times V}{W}$$
where q is the adsorption capacity (mg g^−1^); Co is the initial dye concentration (mg L^−1^); Ceq is the equilibrium dye concentration (mg L^−1^); V is the volume of the solution (L); W is the weight of the sample (g).

#### 2.4.2. Photocatalytic Testing of Samples

Experiments for the photocatalytic degradation of dye were carried out using a Helios-125-II lamp, Sofia, Bulgaria, Quartz lamp power: UV—125 W and IR-175 W; Radiation spectrum: UVA 315–400 nm—60% and UVB 280–315 nm—40%. Before starting the photocatalytic tests, an empty experiment was performed with a pure dye solution exposed to UV light for 2 h. No discoloration of the solution due to direct photolysis of the investigated dye was observed. Photocatalytic experiments with the three composite materials were performed in quartz cuvettes for spectroscopy. The lamp was located axially in the middle in front of the cuvettes, containing composites PA10, PA20, PA30 at a distance of 20 cm. The amount of catalyst for each experiment was 13.3 g L^−1^. The decolorization efficiency (%) of the solution was calculated by Equation (4):(4)$$\mathrm{Decolorization}\;\mathrm{efficiency}\;\left(\%\right)=\frac{\mathrm{Co}-C}{\mathrm{Co}}\times 100=\frac{\mathrm{Ao}-A}{\mathrm{Ao}}\times 1000$$
where Co (mg L^−1^) is the initial dye concentration, C (mg L^−1^) is the dye concentration at a given time t (min), Ao is the absorption at the absorption maximum of the stock solution, and A is the absorption of the solution at a given moment t (min).

### 2.5. Antibacterial Assay of the Composite Material PA30

The antibacterial activity of the composite material PA30 was assessed according to the standard shake flask method (ASTM-E2149-01). The method provides quantitative data for measuring the reduction rate in number of colonies formed, converted to the average colony forming units per milliliter of buffer solution in the flask (CFU mL^−1^). For preparation of *Escherichia coli (E. coli)* and *Staphylococcus aureus (S. aureus)* suspensions, a single colony from the corresponding stock bacterial cultures was used. The bacterial culture inoculation was done overnight in 5 mL sterile nutrient broth (NB) in a 15 mL sterile falcon and incubated at 37 °C at shaking (230 rpm). The inoculated bacterial culture was diluted with sterile buffer (0.3 mM KH_2_PO_4_) until solution absorbance at 475 nm reached 0.28 ± 0.01 (corresponding to 1.5 ÷ 3.0 × 10^11^ CFU mL^−1^). A final concentration of 1.5 ÷ 3.0 × 10^8^ CFU mL^−1^ (working bacterial suspension) was obtained by dilution into sterile buffer. Thereafter, the fabric (0.03 g) was incubated with 5 mL of bacteria at 37 °C and 230 rpm. For determination of the inoculum cell density the suspensions were withdrawn before introducing the samples and after 1, 2, and 3 h of contact with the materials. Those suspensions were serially diluted in sterile buffer solution, plated on a plate count agar, and further incubated at 37 °C for 24 h to determine the number of surviving bacteria. The antimicrobial activity is reported in terms of Log_10_ (CFU mL^−1^).

## 3. Results and Discussion

### 3.1. Gravimetric Analysis

The fraction of the obtained gel and the weight gain after modification were calculated using Equations (1) and (2), respectively. The results are presented in Table 1. It was observed that an increase in the amount of Zn^2+^ for treating the fabric led to a lower yield of the gel obtained, but the fabric weight gain was greater, probably due to the larger amount of zinc oxide particles. Similar results were obtained in our previous study on the modification of cotton fabric [9].

### 3.2. Characterization by SEM and TEM

The composite materials were examined by SEM to determine the size, shape and distribution of the synthesized nanoparticles in the structure of hydrogel deposited onto the surface of the polyamide fabric. Figure 2 shows the micrographs of the pristine polyamide fabric and PA30 composite material at different magnifications. The surface of the fibers of PA30 material is covered with a layer of the hydrogel, bonding and uniting the individual fibers together. Spherically shaped zinc oxide micro-sized particles are distributed relatively evenly over the fabric surface. They have a flower-like structure that is visible at a higher magnification, but besides this, there are many other zinc oxide particles of granular shape and nanoscale particles included in the hydrogel. The variety of structures is probably due to the rapid nucleation process, which then further proceeds to aggregation and building of different hierarchical formations. Other studies report the form and size of zinc oxide particles to be dependent on the conditions of production, such as the concentration of zinc ions and counter ions, the solvent, the temperature, and on the pH of the medium during the treatment [19]. Flower-like particles have shown improved photocatalytic properties over other nanostructures of rod, leaf, or granular form, due to their open porous structure, which eases the diffusion and transport of dye molecules and oxygen containing particles during adsorption and photoreaction [20,21].

Figure 3 shows TEM images of PA30 and a selected area electron diffraction (SAED) pattern of zinc oxide particles. An uneven hydrogel layer is visible on the fiber surface (Figure 3a), and zinc oxide particles distributed in the hydrogel structure (Figure 3b). The presence of aggregates on the fiber surface is also seen (Figure 3c). The existence of diffusion circles and low-intensity light points in the SAED pattern shows that zinc oxide particles are polycrystalline (Figure 3d).

### 3.3. Investigations on Composite Material PA30 as an Adsorbent for Drimaren Rot K-7B Dye

The effect of different reaction parameters on the adsorption of Drimaren Rot K-7B dye onto PA30 composite material was investigated. A series of experiments performed for the purpose involved varying the following parameters, while maintaining the remaining constants: 1. Adsorbent amount; 2. Initial dye concentration; 3. Temperature; 4. Time of adsorbent/ adsorbate contact.

#### 3.3.1. The Effect of Adsorbent Amount

The conditions of these experiments were 20 mL dye solution at a concentration of 60 mg L^−1^, stored in a dark place at 25 °C, with a contact time of 24 h. The adsorption capacity was calculated by Equation (3), and the degree of decolorization of the solution was calculated by Equation (4). Figure 4a shows an initial increase in the adsorption capacity and its further decrease with increasing the adsorbent dose. The reason for this is probably in the fact that, all free active zones in the adsorption process that increased with the amount of adsorbent remain empty. However, the degree of discoloration of the solution gets higher with the adsorbent dose. Figure 4b shows that 60% of the solution is decolored when adding about 4 g L^−1^ adsorbent. Any further addition of the adsorbent does not cause much significant change in the discoloration efficiency.

#### 3.3.2. The Effect of Dye Concentration

The adsorption capacity of PA30 was investigated by altering the dye concentration in the 10 ÷ 160 mg L^−1^ range at 25 °C. The volume of the test solutions was 20 mL, and the amount of adsorbent was 0.7 g L^−1^. The samples were stored in the dark. The dye concentrations were measured before adding the adsorbent and 24 h later.

As seen from Figure 5a, the adsorption capacity increases with increasing the initial dye concentration and reaches 19 mg g^−1^ at the dye concentration of 160 mg L^−1^. The degree of discoloration of the solution is the highest (26%) at the lowest dye concentration. As solution concentration increased, the discoloration dropped sharply to 11% and was only about 7% at 160 mg L^−1^ (Figure 5b). That indicated the adsorption dependence on the initial concentration of the dye solution.

At a lower dye concentration, the ratio between the number of dye molecules and the adsorption surface is small, hence the adsorption does not depend on the initial concentration. At a higher concentration, more dye molecules come into contact with the adsorption centers and therefore a larger number of molecules are adsorbed. Therefore, the adsorption capacity is increasing, but the degree of discoloration is lower.

#### 3.3.3. Adsorption Isotherms

The adsorption isotherm shows the distribution of the adsorbed molecules between the liquid and solid phases at an equilibrium in the adsorption process. In order to evaluate the adsorption efficiency and its most suitable conditions, it is important to determine the type of adsorption isotherm that describes best the process. In the study, the Langmuir and Freundlich isotherms have been used, and the obtained constants are presented in Table 2.

Langmuir isotherm

The Langmuir Equation (5) is suitable for describing the coverage of the adsorbent surface with a monomolecular layer without interaction between adsorbed molecules.
q_e_ = (q_max_K_L_C_e_)/(1 + K_L_C_e_)(5)

Equation (6) is the linear form of Equation (5) obtained after the necessary transformation.
C_e_/q_e_ = 1/q_max_K_L_ + C_e_/q_max_(6)
where q_max_ is the maximum adsorption capacity of the adsorbent (mg g^−1^), q_e_ (mg g^−1^) and C_e_ (mg L^−1^) are the equilibrium concentrations of the dye in the solid and liquid phases respectively. K_L_ is the Langmuir constant associated with adsorption energy. The q_max_ and K_L_ constants can be determined by the slope and intercept, respectively, of linear plot of C_e_/q_e_ against C_e_ (Appendix A). Their values are presented in Table 2. From the Langmuir isotherm, the dimensionless parameter R_L_ can be calculated, which is determined by the Equation (7) [22,23]:R_L_ = 1/(1+ K_L_C_0_)(7)

The value of R_L_ indicates the pattern of the isotherms and type of adsorption defined as: favorable, unfavorable, linear, and irreversible. The isotherm is favorable, if the adsorption capacity increases rapidly with the concentration of the liquid phase 0 < R_L_ < 1. At R_L_ = 1 the isotherm is linear. At R_L_ > 1 isotherm is unfavorable, and at R_L_ = 0 it does not change, i.e., the amount adsorbed does not depend on the change in concentration and isotherm is irreversible. The calculated R_L_ value is in the 0.9 to 0.3 range at an initial concentration of 10 ÷ 160 mg L^−1^ dye, indicating that its adsorption on the composite material was favorable.

Freundlich isotherm

The Freundlich model for assessment of the adsorption is described by the following Equation (8) [24]:q_e_ = K_F_C_e_^(1/n)^(8)

Its linear form is [25]:Lnq_e_ = LnK_F_ + 1/nLnC_e_(9)

The dependence of Lnq_e_ on LnC_e_ is presented in Appendix A. K_F_ is the relative adsorption capacity (mg g^−1^), and n is the heterogeneous factor (adsorption intensity), and they are determined by the intercept and the slope of the line, respectively. Linear regression analysis has been used to determine the isotherm that describes better the adsorption of Drimaren Rot K-7B on the composite material. The regression coefficient R^2^ indicates that the Freundlich model describes the process better. The found constants of both models and R^2^ values are presented in Table 2.

#### 3.3.4. Temperature Effect

The effect of temperature on the adsorption of the dye onto the composite material was also studied. Figure 6 shows the discoloration of the Drimaren Rot K-7B solution at a concentration of 50 mg L^−1^ with 0.5 g L^−1^ adsorbent at different temperatures. As seen, upon treatment for 300 min at higher temperature at 298 K, 308 K, and 323 K, respectively, the discoloration achieved was 4.4%, 6.5%, and 9.1%, which means that the adsorption process was endothermic. With the temperature increase, various changes in the adsorption system might occur: (a) acceleration of the diffusion of the dye molecules through the outer boundary layer and the inner pores of the adsorbent; (b) enlarging the porosity and total pore volume of the adsorbent over time; (c) an increase in the number of active adsorption centres [26,27].

#### 3.3.5. Kinetic Models and Mechanism

Kinetic studies provide useful information about the efficiency of the adsorption process and its applicability. Several kinetic models are being used to describe adsorption kinetics. The most common are the pseudo-first-order, pseudo-second-order, and the diffusion model inside the particles [28].

Pseudo-first-order

The linear form of the pseudo-first-order kinetic model is represented by the following equation [29]:log(q_e_ − q_t_) = log(q_e_) – K_1_t/2.303(10)
where q_e_ and q_t_ are respectively the amount of dye adsorbed on the adsorbent at equilibrium and at time t. K_1_ (min^−1^) is the adsorption rate constant and is calculated from the slope of the line obtained from the dependence log(q_e_ − q_t_) vs. t.

This kinetic model was applied to study the temperature effect upon adsorption. The results obtained at 298 K, 303 K and 323 K are presented in Appendix A, and the calculated parameters are presented in Table 3.

Pseudo-second-order

The pseudo-second-order kinetic model is discribed by the following equation:t/q_t_ = 1/(K_2_q_e_^2^) + t/qe(11)
where K_2_ is the rate constant (g mg^−1^ min^−1^). The linear plot of t/q_t_ with respect to t at different temperatures is presented in Appendix A [30].

The calculated parameters from this kinetic model are also presented in Table 3. It shows that R^2^ has a higher value at all three temperatures, which indicates the particular kinetic model to be the most appropriate.

Model of kinetic diffusion of the dye adsorption onto the composite material

The adsorption of the solute from the adsorbent involves four main stages. The first involves the diffusion of the solute into the solution to the boundary layer of the solvent around the adsorbent. This stage is not speed determining, if the system is under stirring. At the second step, the solute passes through a hypothetical solvent layer at the adsorbent boundary. At the third step, the solute diffuses into the pores of the adsorbent to the active adsorption centers. In the fourth stage, the adsorbate binds to the active centers in the pores of the adsorbent in a very fast process [31].

The diffusion model inside the particles proposed by Weber and Morris was used to identify the adsorption mechanism of the dye from the composite material [32,33].

It is represented by the following equation:q_t_ = K_id_ (t)^0.5^ + C(12)
where q_t_ is the adsorbed dye (mg g^−1^), K_id_ is the diffusion rate constant (mg g^−1^ min^−0.5^), and C is a constant giving information about the thickness of the boundary layer (mg g^−1^).

If the adsorption process follows a diffusion pattern inside the particles, then the dependence should be a straight line passing through the origin of the coordinate system, with a slope K_id_. The plot of q_t_ vs. t^0.5^ is shown in Appendix A at different temperatures. Since all three plot lines do not pass through the beginning of the coordinate system, the diffusion model is not the only speed-determining stage, i.e., the process is complex and involves more than one mechanism. The value of intercept C indicates the thickness of the boundary layer. The higher its value, the greater the effect of the boundary layer is [34]. The values of K_id_, C and R^2^ are presented in Table 3. K_id_ values increase with increasing the temperature, what indicates an acceleration of the dye molecules diffusion from the outer surface of the adsorbent into the macropores. The obtained regression coefficients are in the range 0.93–0.99 and comparable to those obtained in the pseudo-second-order of the reaction, which means that both kinetic models describe the adsorption of the studied dye onto the composite material.

#### 3.3.6. Thermodynamic Adsorption Studies

The thermodynamic parameters (enthalpy ∆H°, entropy ∆S°, and Gibbs free energy ∆G°) for the dye adsorption on PA30 are calculated from the graph obtained from the temperature dependence of the partition coefficient (K_D_).

Equation (13) shows how the partition coefficient is calculated.
K_D_ = q/C_e_(13)
LnK_D_ = ∆S°/R − ∆H°/(RT)(14)

Gibbs free energy is calculated by the equation:∆G° = −RTLnK_D_(15)
where R (0.0083 kJ mol^−1^ K^−1^) is the gas constant, T (K) is the absolute temperature.

Appendix A shows the dependence of LnK_D_ on 1/T. The most probable line was obtained from the slope and intercept of the plot, the values of ∆H° and ∆S° are calculated using Equation (14) and are presented in Table 4 [35,36].

Depending on the type of bonds formed between the adsorbent and the adsorbate, adsorption can be physical or chemical. In the case of weak surface bonds, such as van der Waals or dipole bonds, physical adsorption occurs. The enthalpy of this process is usually less than 20 ÷ 25 kJ mol^−1^. In this case, it is possible several adsorbate layers to be formed over the adsorbent surface. In the case of chemical adsorption, the enthalpy value is 200 ÷ 400 kJ mol^−1^, forming chemical bonds and a single-molecule layer. The calculated standard enthalpy in this study was 19.74 kJ mol^−1^. The enthalpy value and the positive sign indicate that the adsorption is a physical, endothermic process [37].

The Gibbs energy value was calculated by Equation (15) and is presented in Table 4. Gibbs’s energy is an indicator of the spontaneity of the adsorption process. Its positive value at 298 K indicates that the process is spontaneous at this temperature. The entropy is positive, i.e., various structural changes occur during the sorption process, leading to a greater disorder in the system. The hydrogel probably swells over the fibre surface with time, which increases the number of active centers and facilitates the access of the dye molecules to them. The activation energy for dye adsorption was determined by the Arrhenius Equation (16): LnK_2_ = LnA − E_a_/(RT)(16)
where K_2_ is the rate constant (g mg^−1^ min^−1^) in the pseudo-second-order of reaction. E_a_ is the activating energy (kJ mol^−1^), A is the Arrhenius coefficient (pre-exponential factor) related to the total number of impacts of the molecules per second, R (0.0083 kJ mol^−1^ K^−1^) is the gas constant, T (K) is the absolute temperature.

The dependence of LnK_2_ on 1/T is presented in Appendix A. The activation energy of the process is calculated from the slope of the straight line, passing close to three points. Its value is 28.58 kJ mol^−1^ and indicates that, the process proceeds with a relatively low energy barrier, which is characteristic of physical adsorption. The activation energy for the physical adsorption is in the interval 0 ÷ 40 kJ mol^−1^, while for chemical adsorption its value is higher in the range of 40 ÷ 800 kJ mol^−1^ [25,38].

#### 3.3.7. The Effect of the Processing Time on Dye Adsorption

In order to evaluate the possibility of accelerating the discoloration of the dye solution, studies have been carried out on increasing the amount of adsorbent, reducing the volume of the solution and activation of the zinc oxide particles via light illumination. For this purpose, two experiments were performed to discolor a 3 mL dye solution at a concentration of 50 mg L^−1^ and 13.3 g L^−1^ adsorbent in the dark and under UV irradiation.

Figure 7 compares the discoloration of the solution with the time in the dark and upon explosion to UV light. 100% discoloration was achieved in about 200 min under irradiation, while without irradiation, the discoloration achieved for 350 min was 86%. Adsorption in the dark proceeds evenly compared with the one under irradiation. Initially, the rate of the process was high and gradually slowed down due to the decrease of free adsorption centers on the adsorbent surface. Under light illumination, adsorption and photocatalysis processes take place. In this case, reactive oxygen species that attack and decompose the dye molecules have been formed.

#### 3.3.8. Investigations on the Composite Material Polyamide Fabric-Hydrogel-Zinc Oxide Nanoparticles as a Photocatalyst for Decolorization of Dye Aqueous Solutions

Repeated use of the resulting composite materials

All three composites PA10, PA20, PA30 were investigated as reusable photocatalysts. A sample of each material (13.3 g L^−1^) was used three times for 150 min in each treatment and after immersion into a new solution (3 mL) at a dye concentration of 50 mg L^−1^. The obtained results are presented in Figure 8. In the course of the first use, samples PA30 and PA20 decolor faster and more largely the dye solution (94%) than with sample PA10. The second use of all three samples results in a slight delay in the discoloration process, but the most effective is the PA30 sample. Within 150 min, 80% discoloration was achieved. For the third use, samples PA30 and PA20 again discolor the dye solution equally up to 66% for 150 min.

First and second-order kinetic models were used to study the photodegradation kinetics, which are described by Equations (17) and (18) respectively.
Ln(C_0_/C_t_) = K_1_t(17)
1/C_t_ = K_2_t + 1/C_0_(18)
where C_0_ is the initial concentration of the dye solution, C_t_ is the concentration of the dye solution at time t. K_1_ (min^−1^) and K_2_ (mg L^−1^ min^−1^) are the rate constants of the first and second-order kinetic equations. The values of the rate constants for repeated use of composite materials are obtained from the slope of the lines, respectively, from the dependence of In (C_0_/C_t_) on time and the dependence of 1/C_t_ on time. The data obtained are presented in Table 5. The regression coefficients R^2^, calculated after applying the linear regression analysis are the closest to one for the first-order kinetic model in the first and in the second use of the three samples, and are better interpreted by this kinetic model, respectively. For the third use of the samples, the process is better described by the second-order kinetic model, based on the values of the regression coefficients R^2^. This is probably due to a change in the structure of the adsorbent associated with a decrease in the active centers number. The rate constants calculated from the two models as well as the values of R^2^ are presented in Table 5.

Accelerated decolorization of the dye solution when exposed to UV light and upon the addition of hydrogen peroxide

Accelerating of discoloration was investigated in the presence of H_2_O_2_. Its addition to the reaction system probably increases the number of hydroxyl radicals formed, which also enhances the dye molecules degradation. After 90 min, the dye solution discolors 83% in the presence of PA30 alone. The addition of 0.07 mmol L^−1^ and 0.14 mmol L^−1^ H_2_O_2_ resulted in 89% and 95% discoloration, respectively. After 150 min, the difference in the degree of discoloration between the individual solutions remains but is insignificant (Figure 9). A first-order kinetic model describes the reactions. The rate constants of the processes with added H_2_O_2_ and the regression coefficients R^2^ are presented in Table 6.

#### 3.3.9. Antibacterial Activity

The antibacterial effect of the composite material PA30 was evaluated against both *S. aureus* and *E. coli* bacteria. The material was able to eliminate completely *S. aureus* after 2 h of contact (Figure 10a). Two logs reduction of *E. coli* was achieved at the first hour of contact and this effect was maintained within the next two hours (Figure 10b).

In fact, it has been shown that PA30 composite material containing ZnO particles inhibits the growth of both Gram-positive and Gram-negative bacteria, being more effective against the Gram-positive (*S. aureus*) (lower MICs) than against the Gram-negative strains (*E. coli*) (Appendix A).

## 4. Conclusions

A polyamide fabric was modified with polyacrylamide hydrogel and in situ synthesized ZnO particles. The applied method was surface-initiated photopolymerization by visible light. SEM analyses have shown that the obtained larger ZnO particles have a flower-like structure, while the others are smaller with a granular shape. Three materials have been derived by different concentrations of zinc nitrate (10% (sample PA10), 20% (sample PA20), and 30% (sample PA30) of the weight of the fabric, respectively)). In fabric PA30, there is a lower gel yield, but the fabric weight was greater. For this reason, PA30 was studied as an adsorbent of reactive dye Drimaren Rot K-7B and as a photocatalyst. The reaction parameters showed that dye adsorption depends on the concentration of the dye molecules and the active adsorption centers. Thermodynamic study showed that adsorption is a spontaneous and endothermic physical process. The first-order kinetic reaction model describes the process in which the active centers of the composite materials and the amount of the formed highly efficient oxidizing radicals are numerous. After repeated use of the materials, with a decrease in the number of active centers, the process is better described by the second-order kinetic model. PA30 exhibits antibacterial activity against Gram-negative bacteria *E. coli* but effectively eliminates Gram-positive *S. aureus* bacterial cells in their free-floating form.

## Figures and Tables

**Figure 1 materials-15-06649-f001:**
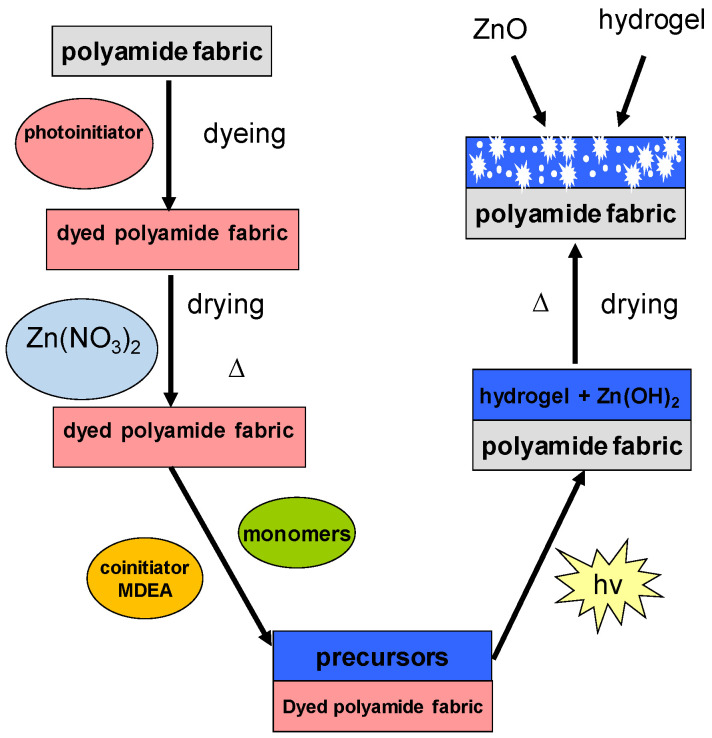
Routes to the preparation of a composite materials ZnO nanoparticles-hydrogel-polyamide fabric.

**Figure 2 materials-15-06649-f002:**
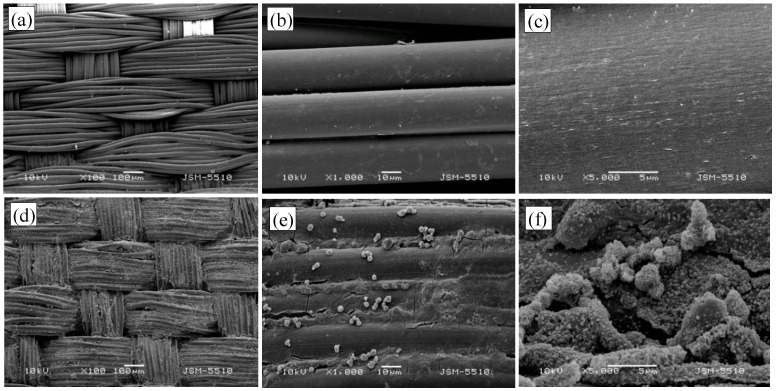
SEM images of the pristine polyamide fabric at different magnification: (**a**) ×100; (**b**) ×1000; (**c**) ×5000 and of composite material PA30: (**d**) ×100; (**e**) ×1000; (**f**) ×5000.

**Figure 3 materials-15-06649-f003:**
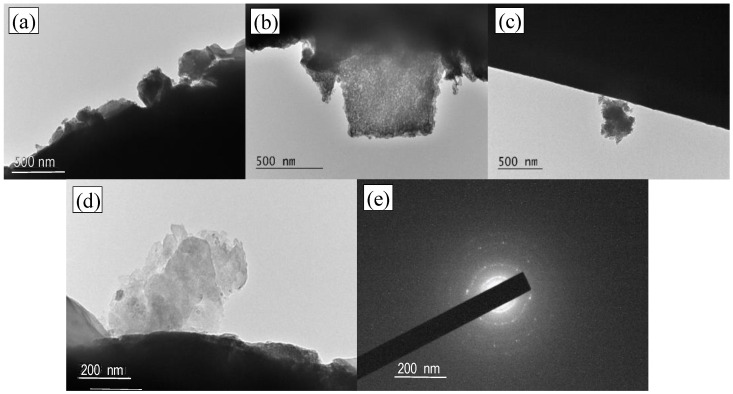
TEM images of PA30 at: a low magnification (**a**–**c**); and at a high magnification (**d**); (**e**) Electron diffraction pattern showing poly rings from zinc oxide nanoparticles.

**Figure 4 materials-15-06649-f004:**
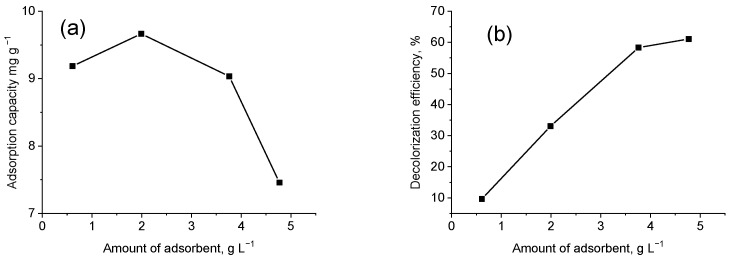
The adsorbent amount effect on: (**a**) adsorption capacity after 24 h; (**b**) discouloration of the solution after 24 h.

**Figure 5 materials-15-06649-f005:**
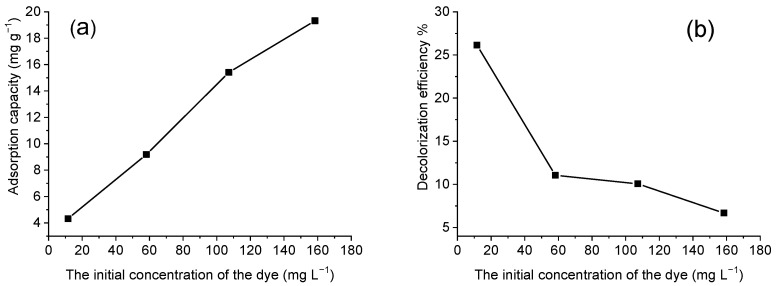
The effect of dye concentration on: (**a**) adsorption capacity and (**b**) discoloration of the solution.

**Figure 6 materials-15-06649-f006:**
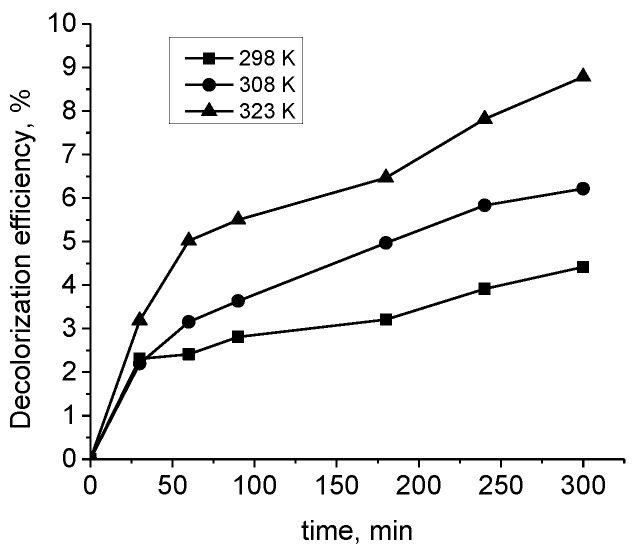
Effect of temperature on the discoloration of the Drimaren Rot K-7B solution at a concentration of 50 mg L^−1^ of 0.5 g L^−1^ adsorbent.

**Figure 7 materials-15-06649-f007:**
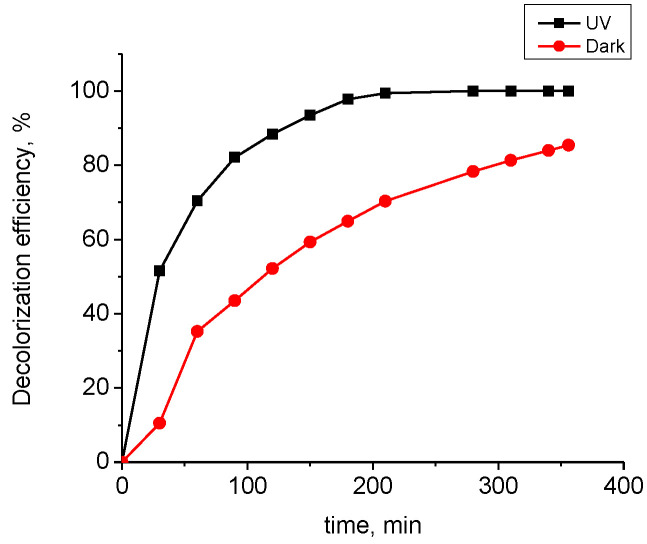
Discoloration in the dark and upon UV illumination of a dye solution at a concentration 50 mg L^−1^ and PA30 composite material (13.3 g L^−1^).

**Figure 8 materials-15-06649-f008:**
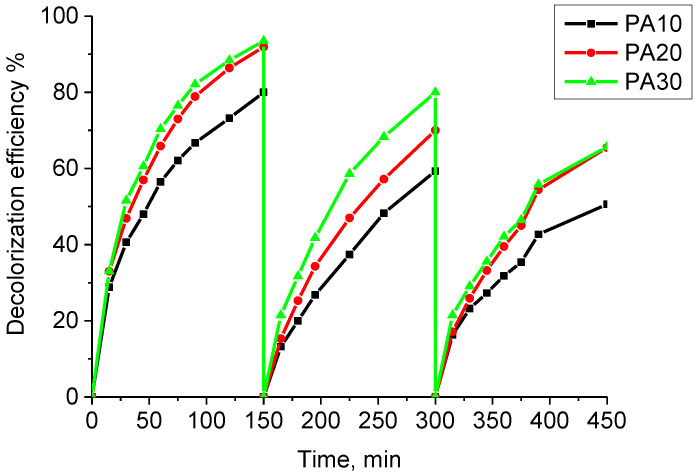
Reuse of PA10, PA20, and PA30 composite materials obtained with UV Light.

**Figure 9 materials-15-06649-f009:**
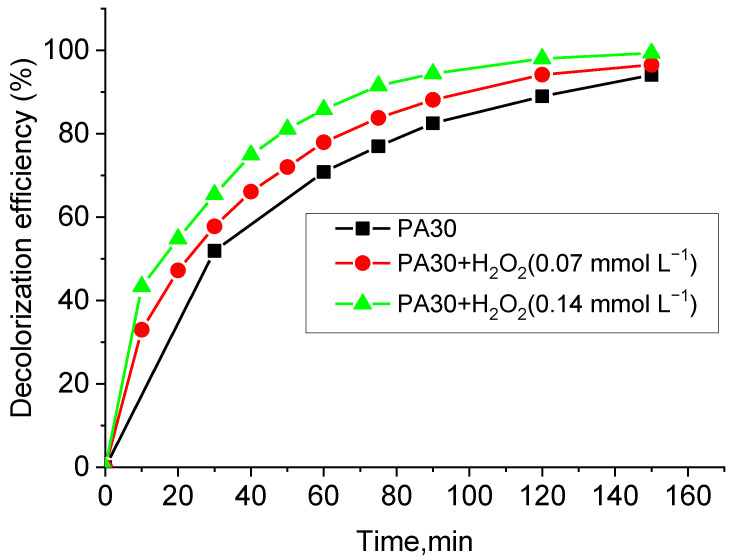
Hydrogen peroxide concentration effect on the decolorization degree of composite material PA30.

**Figure 10 materials-15-06649-f010:**
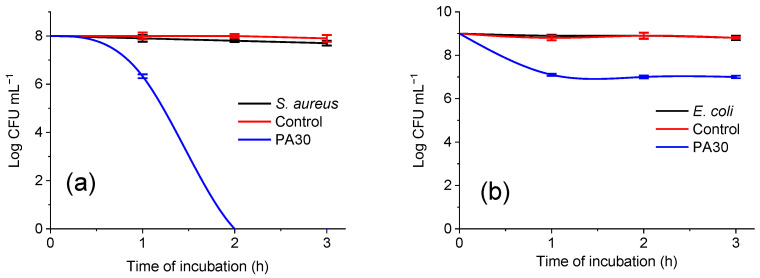
Time killing kinetics of *S. aureus* (**a**) and *E. coli* (**b**) in 0.3 mM KH_2_PO_4_ with cotton fabric, and PA30.

**Table 1 materials-15-06649-t001:** Designation of the samples according to their Zn^2+^ concentration.

Sample	Treatment Conditions, Zn^2+^ (%)	Gel Fraction, (%)	Fabric Weight Gain, (%)
PA10	10	81.1	13.8
PA20	20	75.4	15.9
PA30	30	73.2	21.7

**Table 2 materials-15-06649-t002:** Characteristic coefficients of Langmuir, and Freundlich isotherms (C_0_ = 60 mg L^−1^, 0.7 g L^−1^ adsorbent, 25 °C, 24 h).

Langmuir				Freundlich		
q_m_ (mg g^−1^)	K_L_ (L mg^−1^)	R_L_	R^2^	K_F_ (mg g^−1^)	n	R^2^
26.6	0.015	0.53	0.789	1.030	1.905	0.964

**Table 3 materials-15-06649-t003:** Parameters of the pseudo-first and pseudo-second-order kinetic models and the diffusion model (Weber-Morris) at different temperatures.

Temperature (K)	Pseudo-First-Order			Pseudo-Second-Order			Diffusion Model		
	K_1_ (min^−1^)	q_e_ (mg g^−1^)	R^2^	K_2_ (g mg^−1^ min^−1^)	q_e_ (mg g^−1^)	R^2^	K_id_ (mg g^−1^min^−0.5^)	C (mg g^−1^)	R^2^
298	0.0074	3.38	0.877	0.0042	4.92	0.944	0.1763	1.14	0.936
308	0.0106	6.13	0.956	0.0022	8.00	0.984	0.3573	0.43	0.995
323	0.0080	7.37	0.927	0.0018	10.42	0.962	0.4417	1.28	0.956

**Table 4 materials-15-06649-t004:** Thermodynamic parameters of reactive dye adsorption Drimaren Rot K-7B.

T (K)	LnK_D_	∆S° (kJ mol^−1^ K^−1^)	∆H° (kJ mol^−1^)	∆G° (kJ mol^−1^)	Ea (kJ mol^−1^)
298	−2.40	0.047	19.74	5.936	28.58

**Table 5 materials-15-06649-t005:** Kinetic parameters of PA10, PA20 and PA30 samples in their repeated use.

Number of Uses	Sample	First-Order		Second-Order	
		K_1_	R^2^	K_2_	R^2^
First	PA10	0.00995	0.969	5.14 × 10^−4^	0.982
	PA20	0.01581	0.995	1.37 × 10^−3^	0.883
	PA30	0.01719	0.994	1.71 × 10^−3^	0.867
Second	PA10	0.00585	0.993	1.97 × 10^−4^	0.989
	PA20	0.00777	0.996	3.07 × 10^−4^	0.975
	PA30	0.01039	0.996	5.21 × 10^−4^	0.958
Third	PA10	0.00437	0.943	1.34 × 10^−4^	0.985
	PA20	0.00679	0.987	2.51 × 10^−4^	0.989
	PA30	0.00667	0.973	2.49 × 10^−4^	0.992

**Table 6 materials-15-06649-t006:** Kinetic parameters of the discoloration reaction run with adsorbent PA30 and the addition of H_2_O_2_.

H_2_O_2_ Concentration, mmol L^−1^	K_1_	R^2^	K_2_	R^2^
0.07	0.02356	0.997	0.004	0.790
0.14	0.02937	0.997	0.011	0.734

## Data Availability

Not applicable.

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
