# Peer review of "Design of a Composite Based on Polyamide Fabric-Hydrogel-Zinc Oxide Particles to Act as Adsorbent and Photocatalyst"

_materials, 2022, doi:10.3390/ma15196649_

Round 1

Reviewer 1 Report

The manuscript reports data on developing a composite material as an absorbent and dye removal. The manuscript has not been written properly. The English should be improved. Also, the background of the topic and the novelty of the work are not clear through the manuscript. Some figures are presented without error bars. Authors did not provide enough information about the hydrogel that has been used and its properties.

Author Response

We would like to thank the reviewer for the constructive remarks and comments aimed at improving the quality of our manuscript, which we have tried to address.

The manuscript was revised and the English was corrected. In the introduction, preliminary research is given and the novelty is clarified.  Error bars are also given specifically on the microbiological studies. The hydrogel-based composite material is characterized by TEM, SEM gravimetric analysis.

Reviewer 2 Report

This work is not recommended for publication due to the following concerns. Authors are encouraged to rewrite the manuscript in respect to the following comments.

1. The manuscript is too lengthy and it must be shortened with clear objectives.

2. What is Sct in the title. Typographical errors must be taken care well.

3. Authors should provide convincing experimental evidences to describe the better efficiency of PA30 solid.

4. Characterization of the composite materials is not sufficient. 

5. Some of the contents may be moved to SI.

Author Response

We would like to thank the reviewer for the constructive remarks and comments aimed at improving the quality of our manuscript, which we have tried to address.

1. The manuscript is too lengthy and it must be shortened with clear objectives.

The manuscript has been shortened and the objectives have been formulated.

2. What is Sct in the title. Typographical errors must be taken care well.

The typographical errors were corrected.

3. Authors should provide convincing experimental evidences to describe the better efficiency of PA30 solid.

The explications were done in the text.

4. Characterization of the composite materials is not sufficient. 

We have endeavoured to give a satisfactory characterisation of our composites by SEM, TEM and gravimetric analyses.

5. Some of the contents may be moved to SI.

Some of the Figures were given in the supplementary data.

Reviewer 3 Report

The present manuscript deals with the study of design of a composite based on polyamide fabric-hydrogel- 2 zinc oxide particles to Sct as adsorbent and photocatalyst. The article seems to be interesting, in my opinion it should be revised.

Comments,

1) The experimental results should be discussed in the abstract.

2) Novelty of the work is not clear.

3) In the introduction section, recent works should be discussed and cited. ZnO applications should be mentioned, ref. Journal of Molecular Liquids 322, 114552, 2021; Chemosphere 287, 132086, 2022.

4) 2.2. Preparation of composite materials ZnO nanoparticles-hydrogel-polyamide fabric can be given as a flowchart.

5) Error bars should be included in all the plots.

6) Figures quality should be improved. Some of the figures are not visible clearly.

7) Conclusions should be rewritten. 

8) Too many figures, some of the figures can be given as suppl data. 

Author Response

We would like to thank the reviewer for the constructive remarks and comments aimed at improving the quality of our manuscript, which we have tried to address.

The experimental results should be discussed in the abstract.

Done

2) Novelty of the work is not clear.

The novelty is given in the text.

3) In the introduction section, recent works should be discussed and cited. ZnO applications should be mentioned, ref. Journal of Molecular Liquids 322, 114552, 2021; Chemosphere 287, 132086, 2022.

The articles were cited.

4) Preparation of composite materials ZnO nanoparticles-hydrogel-polyamide fabric can be given as a flowchart.

Done

5) Error bars should be included in all the plots.

Some error bars were included in the figures, especially for the microbiological part.

6) Figures quality should be improved. Some of the figures are not visible clearly.

The quality of the figures was improved.

7) Conclusions should be rewritten. 

The conclusion has been rewritten.

8) Too many figures, some of the figures can be given as suppl data. 

Some of the Figures were given in the supplementary data.

Round 2

Reviewer 1 Report

The revised manuscripts was checked. The conclusion now is too long. Please provide some of the given information in Results and discussion part.

Author Response

The conclusion has been revised. 

Reviewer 2 Report

The revised version may now be considered for publication

Author Response

Thank you for the positive decision